# UNeXt: An Efficient Network for the Semantic Segmentation of High-Resolution Remote Sensing Images

**DOI:** 10.3390/s24206655

**Published:** 2024-10-16

**Authors:** Zhanyuan Chang, Mingyu Xu, Yuwen Wei, Jie Lian, Chongming Zhang, Chuanjiang Li

**Affiliations:** College of Information, Mechanical and Electrical Engineering, Shanghai Normal University, Shanghai 200234, China; changzhanyuan@shnu.edu.cn (Z.C.); 1000466347@smail.shnu.edu.cn (M.X.); 1000484328@smail.shnu.edu.cn (Y.W.); lianjie@shnu.edu.cn (J.L.); licj@shnu.edu.cn (C.L.)

**Keywords:** high-resolution remote sensing images, real-time semantic segmentation, convolutional attention, global–local context, transformer

## Abstract

The application of deep neural networks for the semantic segmentation of remote sensing images is a significant research area within the field of the intelligent interpretation of remote sensing data. The semantic segmentation of remote sensing images holds great practical value in urban planning, disaster assessment, the estimation of carbon sinks, and other related fields. With the continuous advancement of remote sensing technology, the spatial resolution of remote sensing images is gradually increasing. This increase in resolution brings about challenges such as significant changes in the scale of ground objects, redundant information, and irregular shapes within remote sensing images. Current methods leverage Transformers to capture global long-range dependencies. However, the use of Transformers introduces higher computational complexity and is prone to losing local details. In this paper, we propose UNeXt (UNet+ConvNeXt+Transformer), a real-time semantic segmentation model tailored for high-resolution remote sensing images. To achieve efficient segmentation, UNeXt uses the lightweight ConvNeXt-T as the encoder and a lightweight decoder, Transnext, which combines a Transformer and CNN (Convolutional Neural Networks) to capture global information while avoiding the loss of local details. Furthermore, in order to more effectively utilize spatial and channel information, we propose a SCFB (SC Feature Fuse Block) to reduce computational complexity while enhancing the model’s recognition of complex scenes. A series of ablation experiments and comprehensive comparative experiments demonstrate that our method not only runs faster than state-of-the-art (SOTA) lightweight models but also achieves higher accuracy. Specifically, our proposed UNeXt achieves 85.2% and 82.9% mIoUs on the Vaihingen and Gaofen5 (GID5) datasets, respectively, while maintaining 97 fps for 512 × 512 inputs on a single NVIDIA GTX 4090 GPU, outperforming other SOTA methods.

## 1. Introduction

Remote sensing semantic segmentation involves the pixel-level classification of remote sensing images, where each pixel is assigned to a specific semantic category. The objective is to recognize and extract various features or categories of features in remote sensing images, such as buildings, roads, and vegetation [1]. Remote sensing semantic segmentation holds important value and applications. First, it can provide extensive, high-resolution information about land features [2], offering data support for urban planning [3], environmental monitoring [4], resource management [5], and other fields. Additionally, remote sensing semantic segmentation can aid in precise monitoring and management in areas such as agriculture [6], forestry [7], and environmental protection [8]. Moreover, remote sensing semantic segmentation is also crucial for areas such as autonomous driving, intelligent transportation, disaster response [9].

Advancements in sensors have enabled some commercial satellites and aerial remote sensing systems to provide sub-meter resolution [10]. High-resolution remote sensing images can offer highly detailed surface information, capturing internal structures, textures, colors, and other detailed features of the targets [11]. Nevertheless, high-resolution remote sensing images exhibit large intra-class variance and small inter-class variance, with significantly large-scale and complex scenes, requiring extensive computational and storage resources for processing and analysis [12]. This presents challenges to the precision of remote sensing semantic segmentation.

A CNN, as the foundational architecture for many image processing and computer vision tasks, offers robust multi-scale feature extraction capabilities for remote sensing image semantic segmentation. A representative example is the Fully Convolutional Network [13] (FCN): FCNs, by replacing fully connected layers with convolutional layers, extend traditional CNNs, allowing for the processing of variable-sized input images and generating dense predictions for semantic segmentation. U-Net [14]: U-Net features a symmetric encoder–decoder pathway with skip connections to preserve spatial details. It has several variants, such as UNet++ [15], DenseUNet [16], and ResUNet [17], which further improve the accuracy of remote sensing image semantic segmentation. The CNN has many advantages. Due to convolutional operations, a CNN can leverage mechanisms of local connections and weight sharing, greatly reducing the number of parameters, thus reducing model complexity and computational costs [18]. This makes CNNs more efficient in processing high-resolution remote sensing images. Furthermore, a CNN retains more local context information, indicating its powerful segmentation capability for small target objects. However, a CNN, due to the limited receptive field of convolutional kernels, has the drawback of being unable to model global context information, leading to poor performance in semantic segmentation tasks that require long-range information. In order to overcome the limitations of CNNs, researchers have proposed numerous improvement methods. Some researches have expanded the receptive field of CNNs: DeepLabV3 [19] and PSPNet [20] have introduced dilated convolutions and pyramid pooling modules, respectively. Some researches have integrated attention mechanisms into the network, such as MAnet [21] and DANet [22], allowing the model to focus on more important features and enhance overall modeling capabilities. Some researches have designed multi-scale feature fusion networks, such as the FPN [23] (Feature Pyramid Network) and HRNet [24] (High-Resolution Network), to extract and fuse features at different scales, enhancing the model’s multi-scale perception capabilities. Although these models have improved the extraction of important features, they are still limited by the finite receptive field of convolutional kernels and are unable to model global contextual information.

The recent introduction of the ViT (Vision Transformer) appears to address the limitations of CNNs [25]. With a self-attention mechanism, a ViT can model long-range dependencies and exhibit outstanding performance in tasks such as image classification, segmentation, and object detection. The remarkable performance of the ViT has piqued the interest of researchers in the remote sensing field, leading to numerous studies enhancing model accuracy by integrating Transformers. TansUnet [26] combines the encoder–decoder structure of U-Net with Transformer modules to capture both local and global features. UGTransformer [27] integrates Transformers into the decoder to combine with locally extracted features from a CNN, extracting global spatial features for precise segmentation. ST-Unet [28] constructs a decoder with structures parallel to those of a CNN and Swin Transformer, allowing the model to extract more prominent features. HAFNet [29] uses a hybrid attention fusion embedding Transformer, aiming to address the limitations of convolutional operations in capturing global contextual information, thus improving the accuracy of the semantic segmentation of remote sensing images. However, the emergence of global attention can effectively model global contextual information; the stacking of self-attention introduces extremely high computational complexity [30], requiring a substantial amount of computing resources to model global information. This reduces the efficiency of semantic segmentation models for high-resolution remote sensing image processing and limits the potential for further model development.

While CNN-based architectures have achieved significant success in remote sensing image segmentation, they face inherent challenges due to their limited receptive field, which restricts their ability to capture long-range dependencies and global context. Methods such as U-Net and its variants can effectively preserve local details through skip connections, but they struggle with modeling large-scale spatial relationships. On the other hand, Vision Transformers (ViTs) and other Transformer-based models excel at capturing global information due to their self-attention mechanism, but they come with high computational complexity, making them less efficient, especially when processing high-resolution images. This complexity is particularly problematic for real-time applications or environments with limited computational resources. Moreover, although Transformer models perform well in extracting global features, they often struggle to retain local detail, leading to challenges when segmenting small or elongated objects (such as rivers and roads). Existing methods find it difficult to maintain sufficient local information while obtaining global context, resulting in limitations when segmenting small, irregular, or complex-shaped objects.

In this study, we utilized the Vaihingen and GID-5 datasets to evaluate the performance of our proposed method. The Vaihingen dataset includes 33 high-resolution aerial images covering an area of 1.38 km^2^, annotated with six land cover types. This dataset features complex objects such as small cars and visually similar categories like low vegetation versus grass, and it has been widely used in remote sensing and image segmentation research [27,31]. Similarly, the GID-5 dataset provides high-resolution land cover classification across five categories: buildings, farmland, forests, grasslands, and water bodies. It includes pixel-level annotations for 150 scenes and poses challenges in segmenting elongated features such as rivers. This dataset is crucial for assessing the method’s performance in handling various land cover types and complex segmentation tasks [32,33].

Inspired by ConvNeXt [34] and Unetformer, we constructed the UNeXt architecture, which uses ConvNeXt-T as the encoder and builds a lightweight decoder with parallel CNN and Transformer blocks. Our model can learn global and local representations at all stages, and as a result, it achieves excellent segmentation results on the Vaihingen and GID-5 dataset while maintaining an extremely high inference speed.

The primary contributions of this study are as follows:This paper summarizes the existing methods that combine a CNN and ViT for remote sensing image segmentation, uncovering their shared shortcomings, namely, CNN’s inability to model global information and the substantial increase in computational cost brought by VIT. The limited input size of high-resolution remote sensing images constrains global information modeling.To achieve more efficient remote information interaction, we devised a novel and efficient feature fusion module, the SFFB (SC Fuse Block), which incorporates spatial and channel reconstruction convolutions, consisting of an SRU (Spatial Reconstruction Unit) and CRU (Channel Reconstruction Unit). The SRU reduces spatial redundancy through the separate–reconstruct method, while the CRU reduces channel redundancy using the segment–transform–fuse method.To enhance the generalization capability of the Transformer, we designed a novel decoding structure, TransNeXt, which captures multi-scale information through internal multi-scale convolutions and utilizes Spatial Reduction Attention (SRA) to help the Transformer better capture spatial relationships in images, improve the modeling of spatial structures near image boundaries, and effectively improve the segmentation accuracy of elongated objects and object edges while reducing computational costs through channel compression and expansion as well as the combination of residual connections.

## 2. Materials and Methods

### 2.1. Materials

In this study, we evaluated our proposed method through a series of comparative experiments and ablation experiments using the Vaihingen and GID5 datasets, which have different terrain features.

#### 2.1.1. Vaihingen Datasets

The Vaihingen dataset consists of 33 high-resolution aerial images covering a 1.38 square kilometer area with six different land cover classes [31]. These datasets are provided in the form of three-band remote sensing TIFF files (near-infrared, red, green) and single-band DSM, adding complexity and application value to the testing and training of image segmentation models. As shown in Figure 1, it encompasses various urban land cover types such as buildings, trees, cars, low vegetation, and impervious surfaces like streets and sidewalks. The Vaihingen dataset includes small target objects such as cars, and some land cover categories are visually similar, for instance, low vegetation and grass, which pose challenges for the model. In this experiment, we referred to previous works [11,28], after removing mislabeled images, we selected images with IDs 2, 4, 6, 8, 10, 12, 14, 16, 20, 22, 24, 27, 29, and 31 for testing, while the remaining images were used for training. They were then cropped to 512 × 512, and various image augmentation techniques such as random rotations were employed during training. Multi-scale random flipping augmentation was used during testing.

#### 2.1.2. GID-5 Datasets

The GID-5 dataset is a large-scale land cover classification set derived from high-resolution imagery from the Gaofen-2 satellite [32]. As shown in Figure 2, it comprises five primary land cover categories: buildings, cropland, forest, grassland, and water bodies. These aim to provide diverse geographic feature representations for remote sensing tasks. The dataset includes 150 scenes with pixel-level annotations, enabling detailed analysis and model training for segmentation tasks. In the GID-5 dataset, there are numerous elongated rivers, which pose challenges for semantic segmentation models due to their narrow and meandering characteristics. In this experiment, we referred to previous works [32], We used images numbered 1 to 120 for training, and the remaining images numbered 121 to 150 for testing. Similar to the Vaihingen dataset, after cropping them to 512 × 512, various image augmentation techniques such as random rotations were employed during training. Multi-scale random flipping augmentation was used during testing.

### 2.2. Methods

In this section, we first describe the evaluation metrics, network parameters, and experimental setup used in our study. Following this, we present an overview of the UNeXt structure, detailing the design of the employed ConvNeXt encoder structure. Additionally, we discuss two critical modules within UNeXt: the SC Fuse Block and the TransNeXt module.

#### 2.2.1. Implementation Details

Training Settings: All models in the experiments were implemented using the PyTorch framework on a single NVIDIA GTX 4090 GPU. To ensure rapid convergence, the AdamW optimizer was employed for training all models. The base learning rate was set to 6 × $10^{-4}$, with a weight decay of 0.01. A cosine annealing strategy was used to adjust the learning rate. The batch size was set to 8, with a maximum of 80 epochs. All models were trained using an input size of 512 × 512. During testing, MSRF (multi-scale and random flipping) augmentation was applied.

Loss Function: The presence of class imbalance in remote sensing image datasets may cause the model to neglect training on a small number of samples. To mitigate this issue, we introduced additional loss functions, employing both dice loss and cross-entropy loss for model training. The formulas are as follows:(1)$$L=L_{\mathrm{CE}}+L_{\mathrm{Dice}}$$
(2)$$L_{\mathrm{CE}}=-\frac{1}{N}\sum_{n=1}^{N}\sum_{k=1}^{K}y_{k}^{\left(n\right)}\log_{c}{\hat{y}}_{k}^{\left(n\right)}$$
(3)$$L_{\mathrm{Dice}}=1-\frac{2}{N}\sum_{n=1}^{N}\sum_{k=1}^{K}\frac{{\hat{y}}_{k}^{\left(n\right)}y_{k}^{\left(n\right)}}{{\hat{y}}_{k}^{\left(n\right)}+y_{k}^{\left(n\right)}}$$

Evaluation Index: We used the Average Intersection (MioU) and Ave.F1 as the standards to quantify the model’s performance accuracy. These two metrics are derived from the confusion matrix that includes the true positives, false positives, true negatives, and false negatives. The calculation formulas are as follows: $\mathrm{IoU}=\frac{\mathrm{TP}}{\mathrm{FP}+\mathrm{FN}+\mathrm{TP}}$, $F1=2\times \frac{\mathrm{precision}\times \mathrm{recall}}{\mathrm{precision}+\mathrm{recall}}$. In this context, true positive (TP) denotes the number of correctly predicted positive instances, false positive (FP) represents the count of incorrectly predicted positive instances, true negative (TN) signifies the number of correctly predicted negative instances, and false negative (FN) indicates the quantity of incorrectly predicted negative instances. Furthermore, we employ the metrics “number of parameters” (Param.) and “frames per second” (FPS) to assess the computational cost of the model.

#### 2.2.2. Overall Network Structure

The overall structure of UNeXt is depicted in Figure 3. As it is an efficient and precise high-resolution remote sensing image semantic segmentation network, we adhered to the exemplary structure of UNet, encompassing the design of the encoder, decoder, and skip-connection architecture. The encoder and decoder engage in information interaction through skip connections. In particular, UNeXt employs ConvNeXt-T as the encoder and constructs a lightweight decoder, TransNeXt, which runs CNN and Transformer blocks in parallel. Our model is capable of learning global and local representations in all stages, and through the SC Fuse Block, it reduces redundant information in the encoder to more effectively interact with local and global information. The encoder comprises four stages. For an input remote sensing image $X\in R^{H\times W\times 3}$, each stage downsamples the feature map by half. The output of each stage is denoted as $S_{n}$, where *n* = 1, 2, 3, 4. In UNeXt, the feature maps generated by each stage are connected to the SC Fuse Block through 1 × 1 convolution. The output feature map of the nth ConvNeXt block can be represented as $A_{n}\in R^{\left(H/\left(2^{n}\right)\right)\times \left(W/\left(2^{n}\right)\right)\times 2^{n}C_{2}}$, where $C_{2}$ equals 256. After progressing through the four stages of the encoder, we are able to obtain feature maps $F\in R^{(H/32)\times (W/32)\times 1024}$, which are then passed through convolutional layers and input into the decoder. The decoder, similar to the encoder, consists of four stages. At the end of each stage, we use 2 × 2 convolution for upsampling to expand the resolution of the feature maps and reduce the number of channels. The feature *F* is gradually restored to $F^{\prime }\in R^{(H/2)\times (W/2)\times 64}$. Ultimately, by applying 1 × 1 convolution and 3 × 3 depthwise separable convolution to the feature *F*, the original image size is restored, yielding the final prediction result.

#### 2.2.3. Convnext Module

CNN-based encoders such as the ResNet series, such as ResNet-18 and ResNet-50, have played an important role in remote sensing image semantic segmentation in the past. The emergence of VITs has changed the design of semantic segmentation networks for remote sensing images. The global attention mechanism of the VIT brings about a very high level of computational complexity, which can be formulated as follows:(4)$$O(N^{2}\cdot d)$$

In this context, *N* is the length of the input sequence (the number of patches into which the image is segmented). *d* is the feature dimension of each token (the dimension of the embedding vector). Due to the high-resolution nature of remote sensing images, the VIT struggles to handle them. To address this issue, a Swin Transformer adopts a sliding window strategy similar to a CNN, achieving good results as a universal visual backbone in remote sensing image processing. Many advancements of the Swin Transformer in computer vision draw from the strengths of CNN. However, the self-attention mechanism with sliding windows remains complex, leading to a highly intricate network design. On the other hand, the performance advantage of the Swin Transformer often stems from the updated micro-level architectural design, which does not necessarily indicate that CNNs can be completely replaced by Transformers. ConvNeXt inherits the efficient characteristics of Convolutional Neural Networks (CNNs) and, through optimizing updated micro-level convolution operations, can reduce computational complexity while ensuring high performance. This is particularly important for handling large-scale remote sensing image datasets. In contrast to the self-attention mechanism, the powerful modeling capability of convolutions for local information makes convolutional operations perform exceptionally well in segmenting small target objects in remote sensing images. ConvNeXt can provide faster processing speeds while maintaining high accuracy. The design of ConvNeXt is illustrated in Figure 4.

Convnext originates from ResNet-50 and, drawing inspiration from the Swin Transformer, employs depthwise separable convolution for grouping. Subsequently, it replaces the original 3 × 3 convolution kernel with a larger 7 × 7 kernel to expand the receptive field of the CNN. In order to enhance the model’s flexibility and smoothness in handling complex inputs, and to stabilize the training process, Convnext adopts the novel GELU activation function to replace the original ReLU activation function, ultimately yielding the core structure of ConvNeXt. The formula for GELU is GELU(x)=x×P(X⩽x)=x×Φ(x). The symbol Φ(x) denotes the cumulative function of x’s Gaussian normal distribution. Hence, the specific expression of GELU is given by $x\times P\left(X⩽x\right)=x\int_{-\infty }^{x}\frac{e^{-\frac{{\left(X-\mu \right)}^{2}}{2\sigma^{2}}}}{\sqrt{2\pi }\sigma }dX$, where μ and o represent the mean and standard deviation in the normal distribution.

#### 2.2.4. SC Fuse Module

In previous studies on the semantic segmentation of remote sensing images, CNNs effectively extract local features by sliding convolutional kernels over local regions, while Swin Transformers capture global dependencies between arbitrary positions in the input sequence. Combining a CNN and Swin Transformer allows for the extraction of both global and local features. However, to some extent, this method leads to the presence of feature redundancy, as it fails to efficiently utilize the extracted features even when employing a sliding window execution strategy [35]. Additionally, remote sensing images contain dense small-scale terrain information, necessitating the efficient utilization of locally extracted features using a CNN to differentiate small-scale terrain information. Therefore, we propose the SC Fuse Block, which introduces a multi-scale feature fusion mechanism. Through extracting features at different scales, it is possible to better capture objects of different sizes, particularly leading to significant improvements in the detection and segmentation of small targets. Furthermore, using an SRU and CRU to reduce redundant information from both spatial and channel dimensions, the feature representation becomes more compact and effective. This optimization of feature expression and fusion methods allows the model to more efficiently utilize local and global information, thereby improving segmentation accuracy. The components of the SC Fuse Module are illustrated in Figure 5.

The SRU component is depicted in Figure 6. To minimize spatial feature redundancy, the SRU employs separation and reconstruction operations to identify relatively rich spatial features. Specifically, firstly, for an input feature map X=C×H×W, the input *X* is divided into multiple groups using separable convolution, and the mean and standard deviation of each group are used for normalization. The normalized groups are then scaled and shifted using the learnable parameters gamma and beta. Then, the weights of gamma are calculated, and elementwise multiplication is performed on ${\mathrm{gn}}_{x}$ and ${\mathrm{wn}}_{y}$, followed by the application of the sigmoid activation function. Following that, gate thresholds are used to partition the obtained weights into information and non-information masks. The information mask is then multiplied with reweights and input x to yield $x_{1}$, while the non-information mask is multiplied with reweights and input x to yield $x_{2}$. Finally, a standard convolution (pointwise convolution, 1 × 1 convolution) is applied to integrate features from different channels and reconstruct the output.

The CRU component is depicted in Figure 7. For an input feature map X=C×H×W, we apply the split–squeeze–transform–merge operation to diminish channel redundancy.

In particular, we segment the input tensor $X^{w}\in R^{c\times h\times w}$ into several sub-feature maps along the channel dimension. Subsequently, squeezing is applied to split its channel *w* into two parts, resulting in $X_{up}\in R^{c\times h\times a}$ and $X_{low}\in R^{c\times h\times (1-a)}$, to reduce its channel size. Following that, group convolution and pointwise convolution are employed to obtain $Y_{1}$. The result is then concatenated with the input to yield $Y_{2}$. The formulas are as follows: $Y_{1}=M^{G}X_{up}+M^{P_{1}}X_{up}$, $Y_{2}=M^{P_{2}}X_{low}\cup X_{low}$, where “*M*” represents the matrix weights learned by the GWC and PWC. Lastly, adaptive average pooling and softmax are utilized, followed by elementwise multiplication to achieve channel fusion and obtain the final output.

#### 2.2.5. TransNeXt Module

To leverage the global modeling capability of Transformers while preserving local information, numerous prior approaches have integrated convolution and self-attention to build hybrid models. However, the static convolutions they utilize restrict the input dependency of Transformers. Specifically, although the convolutions they employ naturally introduce local information, they have limited impact on improving the model’s representation learning capability. In this study, we introduce a lightweight dual dynamic decoder. TransNeXt constructs two parallel branches to extract global information and local contextual information, dynamically leveraging global and local information, and integrating convolution and self-attention without compromising input dependency. Through incorporating SRA [36] (Selective Region Attention) in the self-attention mechanism, the length and width of the key (K) and value (V) are reduced to maintain feature map resolution and global receptive field while reducing computational complexity. The architecture of the proposed TransNeXt is depicted in Figure 8.

In particular, we partition a feature map $X\in R^{C\times H\times W}$ using a 1 × 1 convolution into two sub-feature maps, denoted as $\left\{X_{1},X_{2}\right\}\in R^{\frac{C}{2}\times H\times W}$. $x_{1}$ and $x_{2}$ are then fed into the global self-attention module of SRA and deformable convolution, producing corresponding feature maps $\left\{X_{1}^{\prime },X_{2}^{\prime }\right\}\in R^{\frac{C}{2}\times H\times \dot{W}}$. These are then concatenated along the channel dimension to yield the feature map $X^{\prime }\in R^{C\times H\times W}$. Effective local token aggregation is accomplished using 1 × 1 convolutions and 3 × 3 depthwise separable convolutions. In general, TransNeXt can be expressed as
(5)Q=Linear(X)
(6)K,V=Split(Linear(Y+LR(Y)))
(7)$$Z=\mathrm{Softmax}(\frac{{\mathrm{QK}}^{T}}{\sqrt{d}}+B)V$$In this context, B represents the relative positional bias matrix that encodes spatial relationships in the attention map, while *d* denotes the number of channels in each attention head. In this context, B represents the relative positional bias matrix that encodes spatial relationships in the attention map, while *d* denotes the number of channels in each attention head.

After the integration of token information at both global and local levels, previous studies have mostly used 1 × 1 convolutions to transform dimensions for channel communication, which has led to computational overhead. Moreover, due to the high similarity of objects in remote sensing images, preserving the original spatial information contributes to distinguishing similar objects. Therefore, this paper proposes a lightweight token enhancer (LTE) that reduces computational costs and achieves lower FLOPs than traditional 1 × 1 convolutions through a combination of channel compression, expansion, and residual connections. As shown in Figure 9, the LTE uses 3 × 3 depthwise convolutions to reduce the number of channels through channel compression, thereby reducing the computational load while preserving crucial information. By introducing residual connections, it retains original information and promotes efficient learning, ensuring optimal feature utilization.

## 3. Experiments and Results

In this section, we evaluated the relative strengths and weaknesses of our method compared to other SOTA (state-of-the-art) methods through comparative experiments and conducted ablation studies to assess the contributions of different components and configurations in the network.

### Comparative Experiments

We conducted comparative experiments by contrasting our proposed UNeXt with several state-of-the-art remote sensing image semantic segmentation networks on the Vaihingen dataset, including MANet, Unetformer, DC-Swin [37], ST-UNet, DeepLabV3+, and ABCNet. It is worth noting that the Vaihingen dataset contains high-resolution urban remote sensing images and is recognized as a challenging dataset for urban environment semantic segmentation. The dataset presents two main challenges:

1. Highly complex urban environment: The dataset contains multiple regions with complex urban environments characterized by a dense distribution of buildings, a mix of various man-made structures and natural elements. The segmentation targets often occlude each other, requiring the handling of complex background samples during the segmentation process.

2. Multi-scale objects, especially small targets: These small targets include vehicles, pedestrians, and small buildings, where detailed shape and information are crucial for the accurate segmentation of remote sensing images.

The results on the Vaihingen dataset: As shown in Table 1, all models were initialized using official pretrained weights. Our approach demonstrates excellent performance. Specifically, UNeXt achieves an mIoU of 84.9% and mF1 of 91.8% on the Vaihingen dataset, benefiting from the preservation of local details using the CNN and the strong global information interaction capability of the Swin Transformer. UNeXt achieves the highest F1 score for buildings, ground, and vehicles. It is noteworthy that the remote sensing semantic segmentation model with a Swin Transformer as the encoder surpasses the segmentation accuracy of the original CNN-based models. This indicates the limitations of a CNN in modeling global information due to its limited receptive field, particularly in scenes with high similarity. However, in improving the original CNN encoder through Convnext, better results are achieved, indicating the potential of CNNs in remote sensing semantic segmentation. Furthermore, CNN-based models perform better in segmenting small objects, emphasizing the importance of local and spatial information in the segmentation of small target objects. Compared to Unetformer, our proposed UNeXt increases in mIoU by 2.2% and mF1 by 2.3%. Compared to other methods, ours performs best in vehicles and ground, demonstrating that UNeXt effectively integrates global and local information, utilizes contextual information effectively, and more accurately segments multi-scale targets.

The visualized segmentation results in the Figure 10 show that in the first row, despite the high similarity between low vegetation and ground, UNeXt can still achieve a more complete and accurate segmentation of highly similar large objects, thanks to the efficient local and global information interaction, using more distinctive features. In the third row, UNeXt accurately segments trees in the low vegetation and tree categories. In the fourth row, compared to other methods, UNeXt accurately distinguishes dense small target objects of the vehicle category, demonstrating that due to the efficient utilization of local information by the CNN, UNeXt performs well for dense small target objects.

The results on GID dataset: The Table 2 presents the outcomes of all approaches within the GID dataset to verify the effectiveness of UNeXt. Our UNeXt achieved 84.2% in MIOU and 89.6% in MF1, surpassing other methods with increases of 1.5% and 2.8% compared to Unetformer, respectively. Because of different data types, the GID dataset contains more roads and rivers, usually appearing in elongated forms. These elongated features, due to their small area relative to the entire image and irregular shapes, make them challenging to accurately detect and classify. MANet and DeepLabV3+ employ attention mechanisms and dilated convolutions, respectively, to expand the receptive field of the CNN. Experimental data suggest that MANet and DeepLabV3+ are still restricted in their ability to interact with global information, and have certain drawbacks in modeling global context, meaning that they are not as effective as our proposed UNeXt. In comparison to other methods, the performance of ST-UNet, which combines CNN and Transformer structures, is not sufficiently outstanding. It is evident that our proposed UNeXt achieved F1 scores of 95.2% and 96.3% for farmland and water bodies, significantly outperforming other models.

The visualization in the Figure 11 shows that rivers, farmland, and grassland are densely distributed in the first row. Because our proposed UNeXt can interact more effectively with local and global information, UNeXt can still achieve relatively accurate segmentation results in such complex scenarios.

## 4. Ablation Study

Influence of Different Encoders: In order to demonstrate that well-designed convolutional models still remain competitive with Transformers, we combined a series of the latest CNN and Transformer backbones with our model for testing on the Vaihingen and GID5 datasets. To ensure fairness, we uniformly cropped input images to a resolution of 512 × 512 and used a consistent batch size of 4 for testing, as indicated in the following Table 3.

Through the utilization of Convnext-T as the backbone network, the highest mIOU and mF1 values were achieved on two datasets, with an mIOU of 84.9% and mF1 of 91.8% on the Vaihingen dataset. When compared with Swin-T, the best-performing Transformer-based backbone network, it resulted in an increase of 0.6% in mIOU and mF1. This demonstrates that, through excellent design, the CNN still possesses significant potential for development in remote sensing semantic segmentation and has not been surpassed by the Transformer. Moreover, owing to the lower computational complexity of CNNs, the improved Convnext-T can segment remote sensing images with relatively fewer parameters and higher precision. Consequently, in the UNeXt model proposed in this paper, Convnext-T was selected as the backbone network.

Each Component of UNeXt: In order to evaluate the performance of the proposed network architecture and the two modules on two datasets with different land features, we conducted a series of ablation experiments on the two datasets with completely identical hyperparameter settings. In this context, the baseline refers to the UNet model with Convnext-T as the backbone network.

Influence of SCFB: The Table 4 indicates that the SCFB can effectively improve the image segmentation performance of the baseline network compared to the baseline network. For instance, when the SCFB is considered in the baseline network, the segmentation structure in Vaihingen shows an increase of 1% in MIOU and of 0.7% in mF1. In GID5, there is an increase of 0.9% in MIoU and of 0.5% in mF1. The SCFB collaborates with the SRU and CRU to reduce redundant information in the CNN, allowing the model to effectively interact with local and global information, thereby improving the utilization of local features and the segmentation accuracy of small target objects. However, it is worth noting that the improvement obtained from using SCFB in GID5 is not as significant as in Vaihingen. This result may be due to the fact that Vaihingen has a larger number of small target objects, such as cars, while GID5 does not have this feature.

As depicted in Figure 12, experimental evidence has shown that the effective reduction in redundant information in CNNs using a SCFB has effectively improved the segmentation accuracy of small target objects.

Influence of Transnext: It is evident that Transnext can effectively enhance the image segmentation performance of the baseline network compared to the baseline network. For instance, when the baseline network considers Transnext, the segmentation structure in Vaihingen shows an increase of 2.6% in MIoU and of 1.8% in MF1. In GID5, there is an increase of 1.7% in MIoU and of 0.5% in MF1. This validates the effectiveness of Transnext in our network. Remote sensing aids the Transformer in better capturing the spatial relationships in the image, improving the modeling of spatial structures near the image boundaries, and effectively enhancing the segmentation accuracy of elongated objects and object edges. As depicted in Figure 13, the visual segmentation results are compared.

Ultimately, by integrating the SCFB and Transnext modules into the baseline network, the overall performance indicators are further enhathe MIoU reached 84.9%, and the MF1 reached 91.8%, increasing by 5.7% and 3.8%, respectively. In GID5, which features a large number of elongated rivers, the MIoU reached 84.2, and the mF1 reached 89.6%, increasing by 2.1% and 1.1%, respectively.

## 5. Complexity Analysis

The Table 5 illustrates the speed and parameter count of various models in Vaihingen and GID5 under the same environment. While the model’s speed may not be the fastest, it outperformed other models on two datasets with different terrain features, achieving an increase of at least 1.8% (MIoU) on Vaihingen and of at least 1.5% (MIoU) on GID5 when compared to the others. It is notable that in terms of speed, models using a Transformer as the visual backbone are slower compared to other models using CNN structures. Evidently, UNeXT achieved good performance while utilizing fewer parameters.

## 6. Results Discussion

The experimental results indicate that ConvNeXt-T, utilized as the backbone network, achieves the highest mIOU and mF1 scores on both the Vaihingen and GID-5 datasets. This demonstrates that CNNs continue to exhibit substantial potential in remote sensing image segmentation, particularly by attaining high segmentation accuracy with lower computational complexity. In contrast, although Transformer-based models excel in global information modeling, they are associated with higher computational costs. The introduction of the SCFB module effectively reduces information redundancy and enhances the utilization of local features, especially yielding more significant improvements on the Vaihingen dataset, which contains numerous small objects. Furthermore, TransNeXt enhances the segmentation precision of boundaries and elongated objects by capturing multi-scale information. Nevertheless, the computational speed limitations of Transformer architectures remain an area that requires further optimization.

## 7. Conclusions

In this paper, we introduce an efficient semantic segmentation model, UNeXt, designed for high-resolution remote sensing images. The structure uses the lightweight ConvNeXt-T as the encoder and a lightweight decoder, Transnext, that combines a Transformer and CNN, along with the SCFB (SCFeature Fuse Block) to integrate the U-shaped encoder. This remote sensing semantic segmentation model can effectively segment high-resolution remote sensing images, reducing computational load while improving the model’s recognition of complex scenes, obtaining global information while avoiding the loss of local details. Specifically, UNeXt can accurately predict two remote sensing semantic segmentation models with different terrain features, enhancing the segmentation effect of elongated objects and features at different scales. Experiments conducted on the ISPRS Vaihingen and GID5 datasets demonstrate that the well-designed CNN structure has not been completely replaced by the Transformer in remote sensing image semantic segmentation tasks and still holds great potential. Additionally, compared to other state-of-the-art methods, UNeXt ensures the light weight of the model while achieving better performance. However, our model still requires further optimization to be practically applicable for processing high-resolution remote sensing images. In future work, we will aim to reduce the model’s computational cost by implementing knowledge distillation, incorporating linear attention mechanisms, and utilizing lightweight Transformer variants to enhance the model’s suitability for real-time applications.

## Figures and Tables

**Figure 1 sensors-24-06655-f001:**
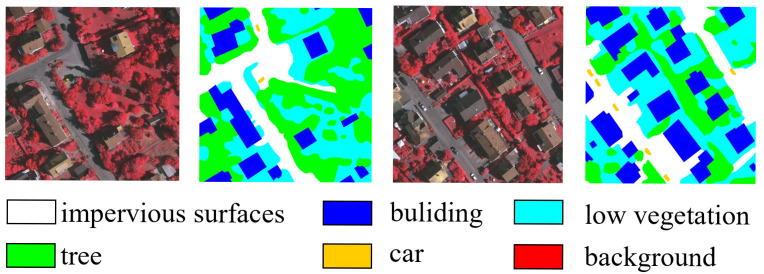
Partial example plot of the Vaihingen dataset.

**Figure 2 sensors-24-06655-f002:**
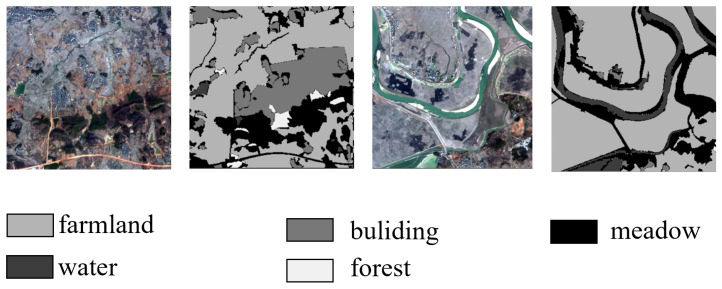
Partial example plot of the GID-5 dataset.

**Figure 3 sensors-24-06655-f003:**
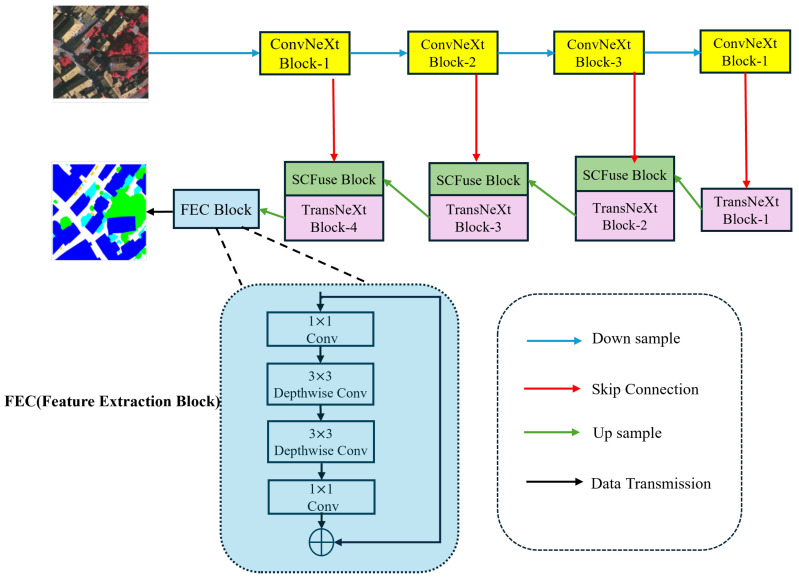
The overall architecture of UNeXt.

**Figure 4 sensors-24-06655-f004:**
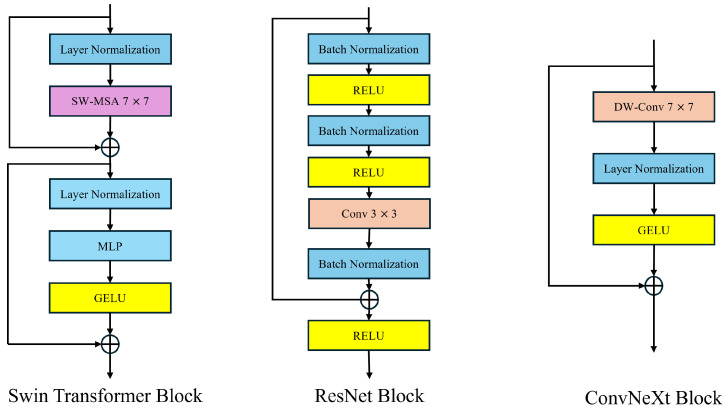
ConvNeXt module structure.

**Figure 5 sensors-24-06655-f005:**
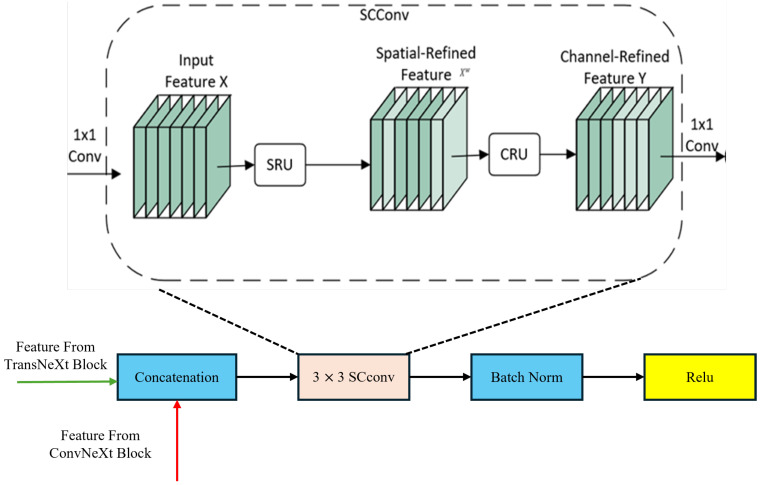
SC Fuse Module structure.

**Figure 6 sensors-24-06655-f006:**
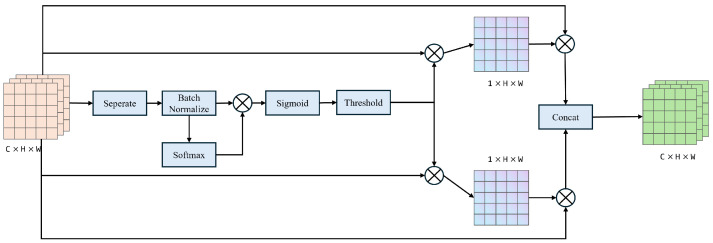
Schematic diagram of SRU component.

**Figure 7 sensors-24-06655-f007:**
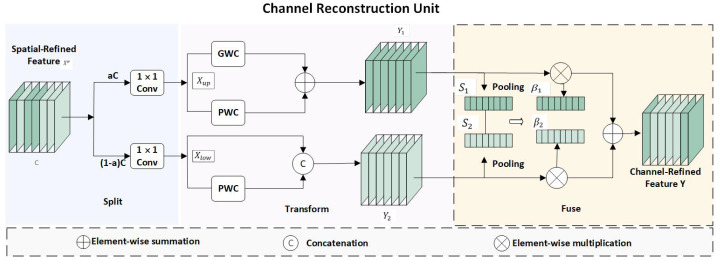
CRU component structure.

**Figure 8 sensors-24-06655-f008:**
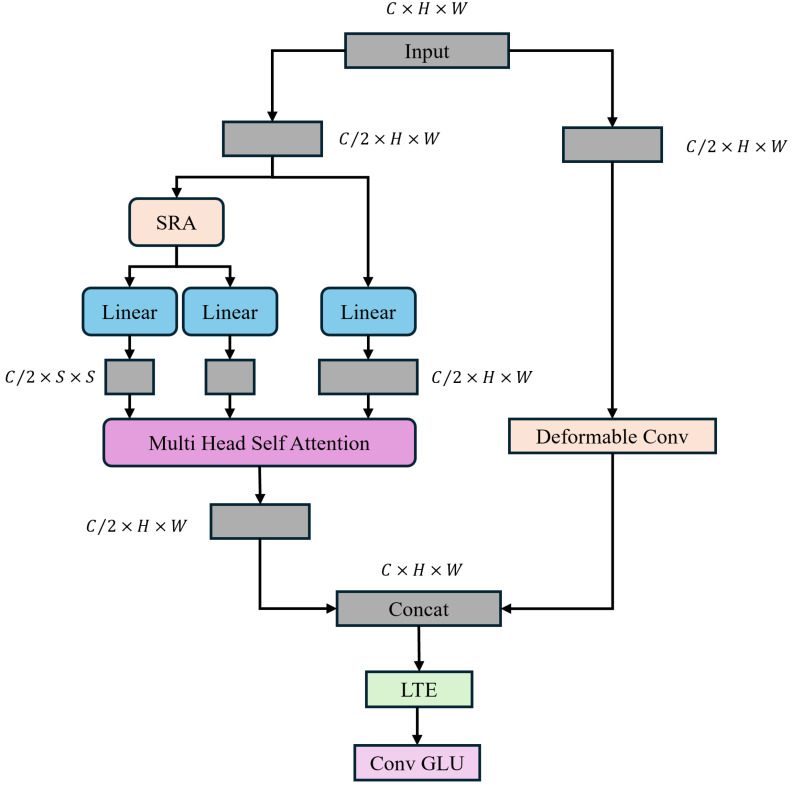
TransNeXt module structure.

**Figure 9 sensors-24-06655-f009:**
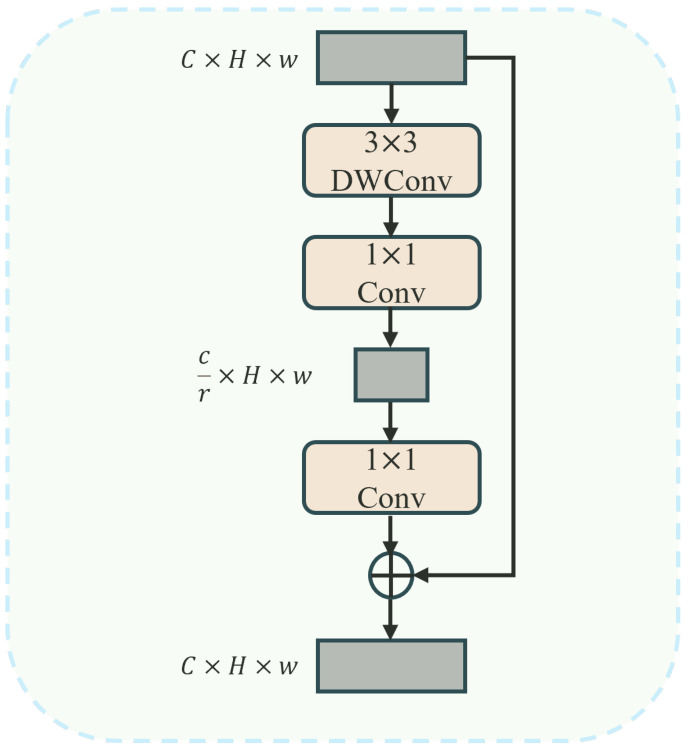
LTE module structure.

**Figure 10 sensors-24-06655-f010:**
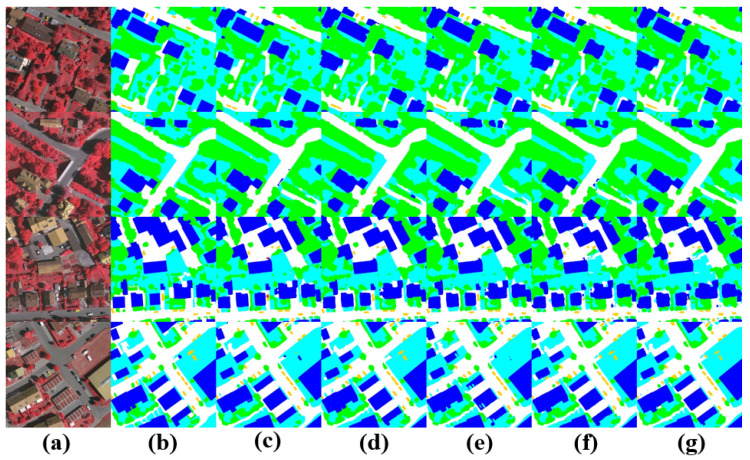
Visual comparison of semantic segmentation for small object features on the Vaihingen datataset. (**a**) Image, (**b**) ground truth, (**c**) UNeXt, (**d**) DeepLabV3+, (**e**) Unetformer, (**f**) ST-UNet, (**g**) DC-Swin.

**Figure 11 sensors-24-06655-f011:**
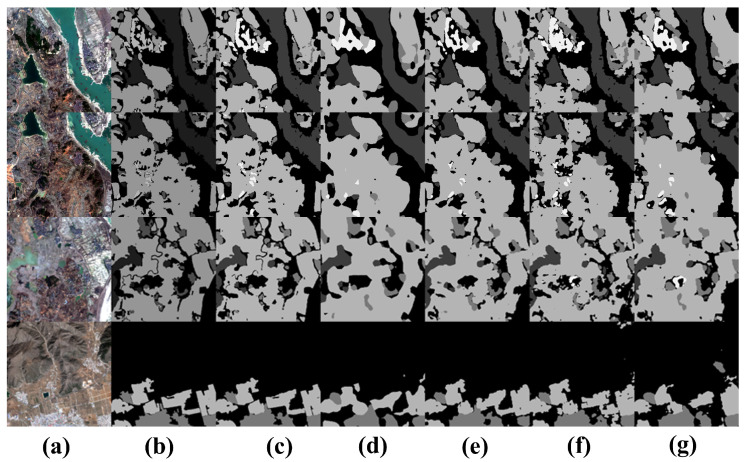
Visual comparison of semantic segmentation for small object features on the GID datataset. (**a**) Image, (**b**) ground truth, (**c**) UNeXt, (**d**) DeepLabV3+, (**e**) Unetformer, (**f**) ST-UNet, (**g**) DC-Swin.

**Figure 12 sensors-24-06655-f012:**
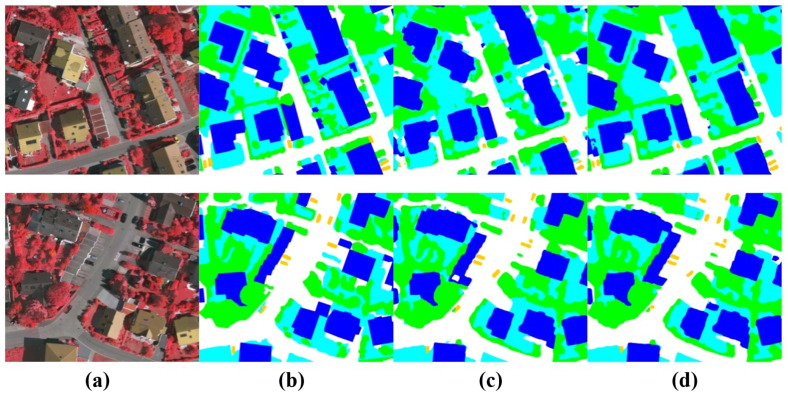
Comparison of segmentation results before and after using a SCFB in the UNeXt framework. (**a**) Image, (**b**) GT, (**c**) UNet, (**d**) UNet+SCFB.

**Figure 13 sensors-24-06655-f013:**
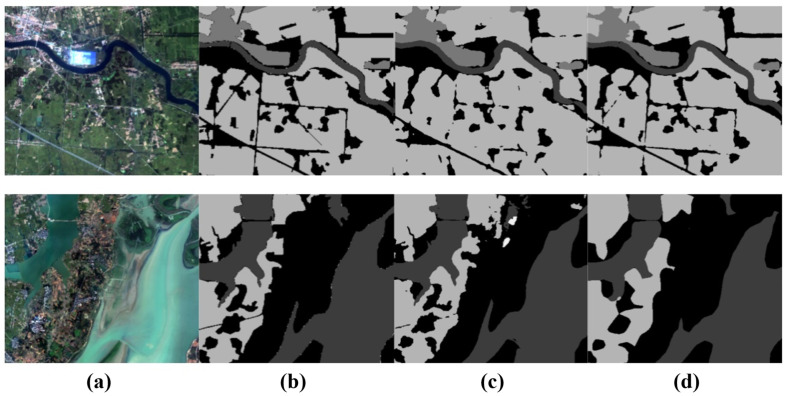
Comparison of segmentation results before and after using Transnext in the UNeXt framework. (**a**) Image, (**b**) GT, (**c**) UNet, (**d**) UNet+Transnext.

**Table 1 sensors-24-06655-t001:** Segmentation accuracy of different methods on the Vaihingen dataset.

Model	Backbone	F1 (%)					Evaluation Index	
		Surface	Building	Low	Tree	Car	MF1	MIoU
MANet	ResNet50	91.4	93.6	82.4	88.5	88.7	90.4	82.7
DeepLabV3+	ResNet50	91.7	94.8	83.0	89.1	88.4	87.9	82.4
ABCNet	ResNet18	92.7	95.2	83.5	89.7	77.7	86.7	77.1
Unetformer	ResNet18	92.7	95.3	84.9	90.6	85.3	89.5	82.7
ST-UNet	/	93.5	96.0	85.6	90.8	86.6	90.3	83.1
DC-Swin	Swin-T	94.1	96.2	85.8	90.4	87.6	90.7	83.2
UNeXt	ConvNeXt-T	96.0	96.2	85.4	90.3	90.8	91.8	84.9

**Table 2 sensors-24-06655-t002:** Segmentation accuracy of different methods on the GID5 dataset.

Model	Backbone	F1 (%)					Evaluation Index	
		Surface	Building	Low	Tree	Car	MF1	MIoU
MANet	ResNet50	92.4	73.9	62.5	88.5	88.7	86.6	82.7
DeepLabV3+	ResNet50	91.6	73.8	63.2	89.1	88.4	87.9	80.1
ABCNet	ResNet18	92.5	74.3	63.4	89.7	77.7	86.8	79.1
Unetformer	ResNet18	93.4	73.4	64.7	90.6	85.3	85.5	82.7
ST-UNet	/	93.1	75.2	63.8	92.8	84.5	87.5	81.1
DC-Swin	Swin-T	94.1	76.2	65.8	93.4	87.2	87.7	81.2
UNeXt	ConvNeXt-T	95.2	75.4	68.4	96.3	84.5	89.6	84.2

**Table 3 sensors-24-06655-t003:** Comparison with state-of-art encoders on the Vaihingen and GID5 datasets.

Model	Backbone	Params (MB)	FLOPs (Gbps)	Vaihingen		GID5	
				MIoU (%)	MF1 (%)	MIoU (%)	MF1 (%)
Transformer-Based	VIL-M	40.5	45.3	83.6	88.9	81.3	89.4
	Focal-T	61.4	74.2	84.1	90.4	82.4	88.2
	DeiT-S	42.7	53.3	83.3	90.9	82.5	89.1
	CSwin-T	41.9	49.6	83.8	91.3	82.1	88.5
	Swin-T	38.2	37.6	84.3	91.2	83.5	87.6
CNN-Based	ResNet 18	23.9	18.5	84.1	89.7	83.6	88.4
	ResNet 50	190.2	100.3	82.7	90.3	82.1	86.2
	VAN-Base	36.5	40.2	84.3	91.4	83.9	87.1
	Convnext-T	38.4	34.2	84.9	91.8	84.2	89.6

**Table 4 sensors-24-06655-t004:** Results of ablation experiments on the Vaihingen and GID5 datasets.

Model	Vaihingen		GID5	
	MIoU (%)	MF1 (%)	MIoU (%)	MF1 (%)
Baseline	79.2	88.0	82.1	88.5
Baseline+SCFB	80.2	88.7	83.0	89.0
Baseline+Transnext	81.8	89.8	83.8	89.0
Baseline+SCFB+Transnext	84.9	91.8	84.2	89.6

**Table 5 sensors-24-06655-t005:** Comparison of model parameters and accuracy.

Model	Params (MB)	Potsdam		Vaihingen	
		Speed (FPS)	MIoU (%)	Speed (FPS)	MIoU (%)
MANet	35.9	90	82.7	91	82.7
DeepLabV3+	41.2	85	82.4	95	80.1
ABCNet	14.0	110	77.1	110	79.1
Unetformer	11.7	135	82.7	135	82.7
ST-UNet	160.97	12	83.1	14	81.1
DC-Swin	45.6	54	83.2	56	81.2
UNeXt	38.4	97	84.9	99	84.2

## Data Availability

The authors would like to thank the International Society for Photogrammetry and Remote Sensing (ISPRS) for providing the Vaihingen and the GID benchmarks. These data can be found at https://www.isprs.org/education/benchmarks/UrbanSemLab/2d-sem-label-vaihingen.aspx and https://x-ytong.github.io/project/GID.html (accessed on 1 October 2024).

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
