# Peer review of "UNeXt: An Efficient Network for the Semantic Segmentation of High-Resolution Remote Sensing Images"

_sensors, 2024, doi:10.3390/s24206655_

Round 1
Reviewer 1 Report
Comments and Suggestions for Authors
I suggest authors to include a paragraph, indicating the limitations of existing work.
I wonder how the high computational complexity of Transformer models can be improved for more efficient use in remote sensing segmentation, as the authors proposed using a Transformer-based framework combined with CNNs to improve performance.
The resolution of Figures 1, 2, 7, and 11 should be improved.
In my opinion, preserving details in high-resolution images is difficult, especially for small and irregular objects, though the authors suggested a hybrid approach to balance local and global details for better accuracy.
I think balancing global and local context is still a challenge with CNN and Transformer integration, even though the authors proposed multi-scale feature extraction to solve this issue.
I fear the computational cost of Transformer models may affect their use in real-time applications, despite the authors claiming their model is more efficient.
I wonder how current methods can improve segmentation of small, elongated objects like rivers and roads, and the authors aimed to improve this with their proposed multi-scale attention mechanism.
In my opinion, the heavy computational load on high-resolution images needs more optimization for practical use, even though the authors said their model performs better than existing methods.
Author Response
Response to reviewer 1
Dear reviewer,
Thank you very much for the valuable comments concerning our manuscript entitled “UNeXt: An Efficient Network for Semantic Segmentation of High-Resolution Remote Sensing Images” (sensors-3136123). We have carefully considered thoughtful suggestions by the reviewers and revised the manuscript accordingly. Please see below for our detailed responses, and we have included a revised version of the original manuscript with all modifications highlighted in red. Additionally, attached to this correspondence is our reply addressing the comments provided by the reviewers. These comments have been reproduced, with our responses provided in blue for clarity. The main corrections in the manuscript and the responses to the reviewers’ comments are as follows.
Reviewer #1:
- I suggest authors to include a paragraph, indicating the limitations of existing work.
Reply: We sincerely appreciate the detailed feedback from the reviewer on our work. Regarding the limitations of existing methods that you mentioned, we have added a discussion in lines 68-81 of the manuscript. Specifically, we point out that while CNNs efficiently extract local features, their limited receptive field constrains their ability to model long-range dependencies and global context. On the other hand, although Transformers excel in capturing global information, their self-attention mechanism introduces higher computational complexity, which is particularly disadvantageous for real-time applications when processing high-resolution images. Additionally, we discuss the limitations of existing methods in handling irregularly shaped objects or small targets, such as rivers and roads. We hope this additional discussion further clarifies the problems our work aims to address, and we sincerely thank you for your valuable comments.
- I wonder how the high computational complexity of Transformer models can be improved for more efficient use in remote sensing segmentation, as the authors proposed using a Transformer-based framework combined with CNNs to improve performance.
Reply: We sincerely thank the reviewer for their attention to our work. We fully acknowledge the issue of computational complexity in Transformer models when processing high-resolution images. The self-attention mechanism in Transformers interacts with every pixel in the input image globally, and the computational complexity of this global self-attention operation is ,where n is the length or width of the input features, resulting in high computational overhead. To address this issue, we have proposed several improvement strategies. First, we introduced the SC Feature Fuse Block (SCFB), which significantly reduces computational complexity by minimizing redundant information in both spatial and channel dimensions. Additionally, UNeXT incorporates Selective Region Attention (SRA) into the self-attention mechanism, selectively reducing the size of the "keys" (K) and "values" (V) in the feature map to lower the computational load of global features. Specifically, SRA reduces the resolution of the image to shorten the length and width of these features, thereby alleviating the computational burden at each step. This optimization is particularly effective in significantly reducing complexity when dealing with high-resolution images.
- The resolution of Figures 1, 2, 7, and 11 should be improved.
Reply: Thank you for your valuable suggestion. We will replace Figures 1, 2, 7, and 11 with higher-resolution versions to ensure better clarity and readability.
- In my opinion, preserving details in high-resolution images is difficult, especially for small and irregular objects, though the authors suggested a hybrid approach to balance local and global details for better accuracy.
Reply: Thank you for your insightful comments. We agree that preserving details in high-resolution images, especially for small and irregular objects, poses significant challenges. To address this issue, our proposed hybrid approach integrates both CNN and Transformer modules, specifically designed to capture local details and global context. The CNN component focuses on retaining fine-grained local information, which is crucial for segmenting small or irregular objects, while the Transformer module captures long-range dependencies to ensure global context is also considered. Our experimental results demonstrate that this approach improves segmentation accuracy, particularly for small and elongated objects such as roads and rivers.
- I think balancing global and local context is still a challenge with CNN and Transformer integration, even though the authors proposed multi-scale feature extraction to solve this issue.
Reply: Your concerns are valid, and we recognize that this issue remains challenging. Although we proposed multi-scale feature extraction and the Selective Region Attention (SRA) mechanism to balance the learning of local and global information, compared to traditional self-attention mechanisms, SRA more effectively integrates local contextual features, preventing the dilution of local information by global features. Additionally, we employed Deformable Convolutions to enhance the model’s ability to handle fine-grained targets. By combining SRA with Deformable Conv for multi-scale feature extraction, selective region attention is computed at different scales. This allows the Transformer to process images across multiple scales, effectively capturing relevant features when dealing with targets of various sizes. This multi-scale processing is especially critical for remote sensing image segmentation, enabling a balance between capturing both macro structures and micro details.
- I fear the computational cost of Transformer models may affect their use in real-time applications, despite the authors claiming their model is more efficient.
Reply: Your concerns are valid. Since the Transformer interacts with every pixel in the input object globally, the Transformer architecture indeed has drawbacks in terms of computational cost. Regarding your concern about the impact of the Transformer model’s computational cost on real-time applications, we have taken this into account in our design. In our work, we addressed this by integrating a lightweight architecture using ConvNeXt-T as the encoder and the TransNeXt module as a lightweight decoder. These optimizations significantly reduce overall computational complexity while still benefiting from the global context modeling of the Transformer. Additionally, the SC Feature Fuse Block (SCFB) helps further streamline processing by minimizing redundancy in both spatial and channel dimensions. Although UNeXT may not be the fastest model, it outperforms other models on two datasets with different terrain features. In the future, we plan to further optimize the computational cost of Transformer models by employing techniques such as knowledge distillation, incorporating linear attention, and adopting lightweight Transformer variants to make these models more suitable for real-time applications.
- I wonder how current methods can improve segmentation of small, elongated objects like rivers and roads, and the authors aimed to improve this with their proposed multi-scale attention mechanism.
Reply: The challenge you mentioned regarding the segmentation of small and elongated objects is precisely one of the issues we aim to address through our multi-scale attention mechanism. By integrating the SRA mechanism into self-attention, UNeXT allows the model to better focus on key local regions while maintaining global awareness. Global context is typically crucial for large structures in remote sensing images, such as urban areas or forests, while local context helps accurately capture small or elongated targets like roads and rivers. SRA scales specific regions, ensuring that global dependencies are modeled while enhancing focus on critical local details. This allows global and local information to work together, improving the segmentation of small and elongated objects like rivers and roads.
- In my opinion, the heavy computational load on high-resolution images needs more optimization for practical use, even though the authors said their model performs better than existing methods.
Reply: We fully agree with your concerns about the computational load of high-resolution images. Although our proposed model performs better than existing methods, we also recognize that further optimization of computational efficiency is needed for practical applications. In the revised manuscript, we have added more discussion on model optimization directions between lines 465 and 470, and proposed potential improvements for future research.
Reviewer 2 Report
Comments and Suggestions for Authors
The authors propose an adapted model for high-resolution image segmentation. The presented model combines UNet + ConvNeXt + Transformer to avoid the loss of local details. The authors also present an ablation study, which reinforces the results obtained. The study is relevant, presents contributions to the area and presents a model with greater precision and shorter execution time. However, I have some suggestions for improving the article:
- The introduction is well explained and justifies the need for the manuscript, but I suggest including a brief paragraph about the data used (Vaihingen and GID-5) along with references that may have also been used;
- In the methodology, the networks and models are well detailed and explained, but I also suggest providing more information about the images used;
- Section 3.1 may fit better into the methodology;
- I believe it would be important to add a discussion section of the results in a separate section.
The article's proposal is relevant, innovative and shows significant results that contribute to the area of remote sensing and artificial intelligence.
Author Response
Response to reviewer 2
Dear reviewer,
Thank you very much for the valuable comments concerning our manuscript entitled “UNeXt: An Efficient Network for Semantic Segmentation of High-Resolution Remote Sensing Images” (sensors-3136123). We have carefully considered thoughtful suggestions by the reviewers and revised the manuscript accordingly. Please see below for our detailed responses, and we have included a revised version of the original manuscript with all modifications highlighted in red. Additionally, attached to this correspondence is our reply addressing the comments provided by the reviewers. These comments have been reproduced, with our responses provided in blue for clarity. The main corrections in the manuscript and the responses to the reviewers’ comments are as follows.
Reviewer #2:
The authors propose an adapted model for high-resolution image segmentation. The presented model combines UNet + ConvNeXt + Transformer to avoid the loss of local details. The authors also present an ablation study, which reinforces the results obtained. The study is relevant, presents contributions to the area and presents a model with greater precision and shorter execution time. However, I have some suggestions for improving the article:
- The introduction is well explained and justifies the need for the manuscript, but I suggest including a brief paragraph about the data used (Vaihingen and GID-5) along with references that may have also been used;
Reply: Thank you for your suggestion. We have added a brief paragraph in the introduction, between lines 82 and 91, introducing the Vaihingen and GID-5 datasets and providing relevant references, so that readers can better understand the sources and characteristics of the data.
- In the methodology, the networks and models are well detailed and explained, but I also suggest providing more information about the images used;
Reply: Thank you for your suggestion. In the methods section, we have further elaborated on the details of the images used. This will help readers better understand our data and experimental environment.
- Section 3.1 may fit better into the methodology;
Reply: Thank you for your insightful feedback on the placement of Section 3.1. After consideration, we have moved the content of Section 3.1 to the methods section to maintain the coherence and logical flow of the paper.
- I believe it would be important to add a discussion section of the results in a separate section.
Reply: Thank you for your valuable suggestion regarding the addition of a separate discussion section. We have added a dedicated discussion section in lines 465-470, where we provide a more in-depth analysis and discussion of the experimental results, including the strengths, limitations, and potential directions for improvement of the model. This will enhance the clarity and depth of the analysis presented in the paper.
The article's proposal is relevant, innovative and shows significant results that contribute to the area of remote sensing and artificial intelligence.
Thank you once again for recognizing our work and for the constructive feedback you provided. We believe that the overall quality of the paper has been significantly improved based on your suggestions.
Round 2
Reviewer 1 Report
Comments and Suggestions for Authors
The authors have responded to my comments. I recommend a thorough review of the technical content and a detailed language proofing of the manuscript to ensure clarity and accuracy.
Author Response
Dear reviewer,
Thank you very much for your time and the valuable comments concerning our manuscript entitled “UNeXt: An Efficient Network for Semantic Segmentation of High-Resolution Remote Sensing Images” (sensors-3136123). We sincerely thank you for your attention to our work.
Best regards,
The authors.
Reviewer 2 Report
Comments and Suggestions for Authors
After reading and analyzing, it is possible to observe that the authors made all the requested corrections. The article is well written, detailed and contributes to the area of remote sensing and artificial intelligence.
Author Response

(The authors gave the same response as above.)
